# Characterisation of miRNA Expression in Dental Pulp Cells during Epigenetically-Driven Reparative Processes

**DOI:** 10.3390/ijms24108631

**Published:** 2023-05-11

**Authors:** Michaela Kearney, Paul R. Cooper, Anthony J. Smith, Henry F. Duncan

**Affiliations:** 1Division of Restorative Dentistry & Periodontology, Dublin Dental University Hospital, Trinity College Dublin, University of Dublin, D02 F859 Dublin, Ireland; michaela.kearney@dental.tcd.ie; 2Department of Oral Sciences, Sir John Walsh Research Institute, Faculty of Dentistry, University of Otago, Dunedin 9016, New Zealand; 3Oral Biology, School of Dentistry, University of Birmingham, Birmingham B5 7EG, UK

**Keywords:** microRNA, epigenetics, RNA sequencing, dental pulp stem cells, non-coding RNA, histone deacetylase inhibitor, DNA methyltransferase inhibitor, vital pulp treatment

## Abstract

Within regenerative endodontics, exciting opportunities exist for the development of next-generation targeted biomaterials that harness epigenetic machinery, including microRNAs (miRNAs), histone acetylation, and DNA methylation, which are used to control pulpitis and to stimulate repair. Although histone deacetylase inhibitors (HDACi) and DNA methyltransferase inhibitors (DNMTi) induce mineralisation in dental pulp cell (DPC) populations, their interaction with miRNAs during DPC mineralisation is not known. Here, small RNA sequencing and bioinformatic analysis were used to establish a miRNA expression profile for mineralising DPCs in culture. Additionally, the effects of a HDACi, suberoylanilide hydroxamic acid (SAHA), and a DNMTi, 5-aza-2′-deoxycytidine (5-AZA-CdR), on miRNA expression, as well as DPC mineralisation and proliferation, were analysed. Both inhibitors increased mineralisation. However, they reduced cell growth. Epigenetically-enhanced mineralisation was accompanied by widespread changes in miRNA expression. Bioinformatic analysis identified many differentially expressed mature miRNAs that were suggested to have roles in mineralisation and stem cell differentiation, including regulation of the Wnt and MAPK pathways. Selected candidate miRNAs were demonstrated by qRT-PCR to be differentially regulated at various time points in mineralising DPC cultures treated with SAHA or 5-AZA-CdR. These data validated the RNA sequencing analysis and highlighted an increased and dynamic interaction between miRNA and epigenetic modifiers during the DPC reparative processes.

## 1. Introduction

Clinical approaches for the management of deep caries and symptomatic pulpitis traditionally involve removal of the entire pulp tissue as part of root canal treatment (RCT), a process which is destructive, costly, and decreases the functional lifespan of the tooth [1]. Given these shortcomings, attention has now shifted towards vital pulp treatment (VPT) procedures, which aim to preserve the pulp and stimulate its natural reparative abilities [2]. Current VPT materials, such as mineral trioxide aggregate (MTA), have demonstrated improved success rates for the treatment of pulpitis compared to calcium hydroxide materials; however, many of these new materials remain limited by their cytotoxicity, non-specific action, poor handling properties, and unpredictable reparative capacity [3].

Exciting opportunities exist for the development of new dental biomaterials, which target the cellular agents regulating dental pulp reparative processes, including epigenetic modulators, such as DNA methyltransferases (DNMTs) and histone deacetylases (HDACs). The therapeutic potential for DNMT inhibitors (DNMTis) has previously been demonstrated, with treatment of human dental pulp cells (DPCs) with the DNMTi 5-aza-2′-deoxycytidine (5-AZA-CdR) resulting in upregulation of a panel of odontogenic markers, such as dentin matrix acidic phosphoprotein 1 (DMP1) and dentin sialophosphoprotein (DSPP), as well as an increase in mineralisation activity [4]. Extensive research has also highlighted the potential therapeutic use of HDAC inhibitors (HDACis) for use in regenerative endodontics. These include the pan-HDACis trichostatin A (TSA), valproic acid (VPA), and suberoylanilide hydroxamic acid (SAHA), among others. In a series of studies, these HDACis were shown to stimulate differentiation and mineralisation in both rodent and human DPC cultures [5,6,7,8], highlighting an exciting avenue for further translational research.

Recently, the role of non-coding RNAs in dental pulp mineralisation and repair has also been highlighted. Specifically, a number of microRNAs (miRNAs) have been identified as being altered in expression during DPC differentiation, such as miR-20a [9], miR-665 [10], and miR-338-3p [11]. Although not recognised as epigenetic modifiers in their own right, the recent literature has highlighted a significant interaction between miRNAs and epigenetic modifications, such as DNA methylation and histone acetylation, in a number of critical cellular processes, including differentiation [12]. Therefore, it is possible that the increase in DPC mineralisation, which occurs following exposure to HDACis [5,6] and DNMTis [4], may be orchestrated by, or related to, miRNA expression. At present, little is known about the nature of these interactions in DPC populations and how they might regulate dental pulp mineralisation and repair. A deeper understanding of this has the potential to lead to the discovery of novel therapeutic targets in the field of regenerative endodontics.

The miRNA expression profile of mineralising human DPCs has previously been analysed using a microarray approach [13]. This study identified 22 differentially regulated miRNAs compared to non-mineralising DPCs after 14 days in culture, with 12 being upregulated and 10 being downregulated. Furthermore, target gene analysis identified genes associated with the Wnt and MAPK signalling pathways, which have been linked to odontoblast-like cell differentiation [14,15]. Although a well established and affordable method for profiling large numbers of miRNAs, microarray analysis lacks the specificity of newer high-throughput methods, such as RNA sequencing (RNAseq), as well as the capacity to detect novel miRNAs and distinguish between variants of existing miRNAs. In addition, while the role of miRNAs, both in DPC mineralisation and as orchestrators of other epigenetic modifications, is a growing area of research, the effect of epigenetic inhibitors on the global miRNA expression profile in mineralising DPCs has not yet been investigated using high-throughput methods.

Here, for the first time, the miRNA expression profile of mineralising DPCs was established using next-generation sequencing, and the effect of two pharmacological epigenetic inhibitors on this profile—a HDACi (SAHA) and a DNMTi (5-AZA-CdR)—was investigated. Furthermore, bioinformatic analyses, such as target gene prediction, pathway analysis, and GO enrichment, were carried out to determine potential mechanisms through which the miRNAs identified might regulate DPC mineralisation, as well as the effects of epigenetic modifications on these mechanisms. Finally, miRNAs were identified from these data, and their expression profiles were validated using qRT-PCR, having the purpose of identifying potential therapeutic targets. The principal aim of this research was to analyse the role of miRNAs in epigenetically-mediated DPC mineralisation, highlight mechanistic interactions, and identify specific miRNAs of interest. A secondary aim was to uncover potential avenues for further investigation, which will ultimately aid in the development of next-generation biologically-based therapeutics for use in regenerative endodontics.

## 2. Results

### 2.1. Growth Rate and Mineralisation in Rodent DPCs

To evaluate the effects of SAHA and 5-AZA-CdR on DPC growth, cells were stained with Trypan Blue (Sigma-Aldrich, Arklow, Ireland) and counted using a haemocytometer (Hausser Scientific, Horsham, PA, USA) in a timecourse study on days 1, 4, 7, 11, and 14 of culture. Four culture conditions were applied: normal medium (standard, non-mineralising), mineralising medium, mineralising medium supplemented with 1 µM SAHA (Sigma-Aldrich, Arklow, Ireland) [6], and mineralising medium supplemented with 1 µM 5-AZA-CdR (Sigma-Aldrich, Arklow, Ireland) [4] (Figure 1). By day 4, DPCs, which had been induced to mineralise (Ind), demonstrated a significant increase in cell growth (2-fold) compared to non-induced (NInd) DPCs and induced DPCs treated with SAHA (Ind+S) or 5-AZA-CdR (Ind+A) (*p* ≤ 0.004). This trend continued through day 7 (*p* ≤ 0.013). By day 11, this increase was significant only in comparison with Ind+A (*p* = 0.023). The positive effect of mineralising medium on DPC proliferation was suppressed by SAHA and 5-AZA-CdR, with cells exposed to SAHA displaying a similar growth pattern to cells in the NInd group, while cells exposed to 5-AZA-CdR demonstrated no discernible increase in cell numbers over the experimental time period, in contrast to other groups (Figure 1).

To confirm induction of mineralisation, DPC cultures were stained with Alizarin Red S (Millipore, Cork, Ireland) on days 11 and 14 (Figure 2).

On day 11, DPCs in Ind+S and Ind+A demonstrated evidence of mineral production, as determined by the presence of red-stained calcific deposits. By day 14, Ind DPCs also exhibited mineral production, an effect that was visibly increased after treatment with SAHA and 5-AZA-CdR. There was no staining evident in NInd DPCs at any point (Figure 2A). As a result, NInd samples were not included in the quantitative comparative analysis, and, instead, the Ind group was selected as the control against which to compare mineralisation in the Ind+S and Ind+A groups. Quantification of Alizarin Red S deposits (Figure 2B) revealed that there was an increase in the Ind+A wells compared to Ind and Ind+S, with this increase being statistically significant compared to Ind+S at day 11 (*p* = 0.033) and compared to both Ind and Ind+S at day 14 (*p* ≤ 0.028). Interestingly, there was a decrease in Alizarin Red S staining in Ind+S compared to Ind at day 11, although an increase was evident at day 14; however, mineralisation per cell analysis using parallel cell counts (Figure 2C) demonstrated an increase in mineralisation produced per cell in Ind+S compared to Ind on both days 11 and 14. A similar effect was seen in Ind+A, with this increase being significant compared to Ind at day 11 (*p* = 0.03), as well as compared to Ind and Ind+S at day 14 (*p* ≤ 0.002).

### 2.2. Small RNA Sequencing and Differential Expression Analysis

In total, 784,172,846 raw reads were obtained from all 12 samples (three replicates for each experimental group). Following alignment and trimming to remove adapter sequences, read length distribution revealed a peak at 22 nucleotides (Figure 3A), corresponding to the length of products of enzymatic digestion by Dicer [16,17]. Based on the known length of small RNAs, sequence reads of 15–35 nucleotides in length were retained for further analysis, resulting in a total of 384,340,259 reads, with an average of 32,003,354 reads per sample. Annotation against tRNAscan-SE [18], Ensembl (Release 98, [19]), and miRbase 21 [20] identified 64,089 sRNAs, of which 4036 were known, and 60,053 were novel. A total of 487 known miRNAs and 11,969 novel miRNAs were identified. Approximately 15–17% of reads in each experimental group were identified as miRNAs. A similar proportion (18–23%) were revealed to be tRNAs, while the remaining reads were classed as small nucleolar RNAs (snoRNAs), small cajal body-specific RNAs (scaRNAs), and small nuclear RNAs (snRNAs). In each group, approximately 50% of reads were classified as unannotated (Figure 3B).

A moderated *t*-test, combined with Storey’s bootstrapping method to correct for false discoveries [21], were used to compare miRNA expression levels between groups, which determined a total of 129 mature miRNAs to be differentially expressed (>1.5-fold, q ≤ 0.05) in four of the five experimental pairings: Ind+S and Ind+A compared to NInd, as well as Ind+S and Ind+A compared to Ind (Table 1). There were no differentially expressed miRNAs which satisfied the fold change cut-off and q-value criteria in Ind compared to NInd. A Mann-Whitney U test was subsequently used on this group to identify differentially expressed miRNAs using a less stringent statistical method. Indeed, 43 differentially expressed miRNAs were identified (Appendix A); however, for the purpose of accurate comparison between groups, these data were not included in subsequent bioinformatic analysis.

Among the other groups, the moderated *t*-test with Storey’s bootstrapping method identified 26 differentially expressed miRNAs in Ind+S compared to NInd, of which 15 were upregulated and 11 were downregulated, with the largest fold increase being 8.5 and the largest fold decrease being 3.2 (Appendix A). There were 30 differentially expressed miRNAs in Ind+A compared to NInd, of which 29 were upregulated, and one was downregulated. Within this group, the largest fold increase was 604, while the largest fold decrease was 1.8 (Appendix A).

Compared to the Ind control, there were 95 differentially regulated miRNAs in Ind+S, of which 62 were upregulated, and 33 were downregulated, with the largest fold increase being 9 and the largest fold decrease being 4.5 (Appendix A). There were 24 differentially regulated miRNAs in Ind+A compared to Ind, of which 23 were upregulated, and one was downregulated. The largest fold increase in this group was 274, while the largest fold decrease was 1.6 (Appendix A).

### 2.3. Functional Annotation

To generate a list of predicted target genes for the differentially expressed miRNAs in each experimental pairing, the in-built target prediction function of Strand NGS ver3.4 (Strand Life Sciences, Bengaluru, India) was used, which utilises the miRDB database (http://mirdb.org/, accessed on 14 October 2020) (miRDB, St. Louis, MO, USA). When compared to the NInd control, the 26 differentially expressed miRNAs in the Ind+S group were predicted to target 150 genes, and the 30 differentially expressed miRNAs in the Ind+A group were predicted to target 468 genes. Compared to the Ind control, 425 genes were predicted to be targeted by the 95 differentially expressed miRNAs in the Ind+S group, while 345 genes were predicted to be regulated by the 24 differentially expressed miRNAs in the Ind+A group. Across all four experimental pairings, a number of target genes were predicted, which have previously been shown to be involved in mineralisation or stem cell processes, including *Sox4*, *Smad4*, *Mmp13*, and *Smad7* (Table 2).

Each list of target genes was uploaded to the Database for Annotation, Visualisation and Integrated Discovery (DAVID) (https://david.ncifcrf.gov/, accessed on 16 October 2020) (Laboratory of Human Retrovirology and Immunoinformatics (LHRI), Frederick, MD, USA) [22,23] for Gene Ontology (GO) enrichment. When compared to NInd, GO analysis highlighted 170 and 434 GO terms in Ind+S and Ind+A, respectively, while in Ind+S compared to Ind, 506 GO terms were identified, and 243 GO terms were identified in Ind+A compared to Ind. Across the four experimental pairings, the target genes were associated with several important processes in mineralisation and stem cell differentiation, such as regulation of bone mineralisation, osteoblast differentiation, biomineral tissue development, tooth mineralisation, stem cell proliferation, and regulation of the MAPK cascade (Table 3). Pathway analysis using the Kyoto Encyclopedia of Genes and Genomes (KEGG) database confirmed these findings, revealing the MAPK, Wnt, and TGF-β signalling pathways to be associated with the target genes of differentially expressed miRNAs (Table 4).

### 2.4. Characterisation of Expression of Selected miRNAs Using qRT-PCR

In-depth analysis of the differentially expressed miRNAs, their potential functions, and previous relevant research enabled the selection of six miRNAs for further analysis using qRT-PCR. Two of these—miR-346 and miR-881-3p—were primarily chosen to validate the RNAseq data, as they had the highest fold change in Ind+S and Ind+A, respectively (Appendix A). Quantification of miRNA expression using qRT-PCR correlated with the findings of the RNAseq analysis, which revealed strong upregulation of miR-346 and miR-881-3p in Ind+S and Ind+A, respectively, on day 4 (Table 5).

The remaining four mature miRNAs—miR-182, miR-200b-3p, miR-221-5p, and miR-205—had predicted target genes, GO terms, and KEGG pathways related to mineralisation or stem cell processes, as well as evidence from the previous literature linking them to mineralisation or epigenetic processes (Appendix A). For example, members of the Wnt family of genes were identified as potential target genes of miR-221-5p, in particular Wnt5a and Wnt11. The Wnt signalling pathway is known to play a critical role in odontoblast differentiation [24]. Target gene analysis also highlighted Dnmt3a as a predicted target gene of miR-200b-3p, which, coupled with results from previous studies showing miR-200b-3p to be regulated by DNMT3A [25], may suggest a negative feedback loop between miR-200b-3p and DNMT3A. The previous literature has also linked miR-182 to mineralisation and epigenetic processes, with one study demonstrating it to be a negative regulator of osteoblast differentiation via FoxO1 [26], a finding which was corroborated by a more recent study showing that downregulation of miR-182 promoted osteoblast proliferation and differentiation in osteoporotic rats [27]. Another study by Inoue et al. [28] revealed that miR-182 played an important role in regulating bone homeostasis [28]. Finally, miR-205 was selected as previous research has suggested links to osteogenesis, with one study identifying it as one of many miRNAs which regulate osteoblast differentiation via Runx2 [29]. This finding was corroborated in a later study, which found that inhibition of miR-205 increased osteogenic capabilities of rodent BMMSCs via upregulation of *Bsp* and *Opn* [30]. Taking these previous findings into account, these four miRNAs were selected for expression analysis on days 1, 4, and 7.

DPCs cultured in mineralising medium demonstrated a clear increase in expression in all four miRNAs by day 4, with this effect being significant in miR-200b-3p (2.7-fold, *p* = 0.002), miR-221-5p (2.8-fold, *p* = 0.013), and miR-205 (9.8-fold, *p* = 0.025). All four miRNAs remained upregulated at day 7, with this finding being significant for miR-221-5p (3.1-fold, *p* = 0.001) and miR-205 (7.5-fold, *p* = 0.001) (Figure 4A).

Treatment with SAHA resulted in an immediate increase in expression in all four miRNAs at day 1 compared to mineralising medium alone, ranging from a 1.6-fold increase in miR-205 (*p* = 0.181) to a 7.1-fold increase in miR-182 (*p* = 0.057). A similar effect was observed upon treatment with 5-AZA-CdR, following which an immediate increase in the expression levels of miR-182 (9.3-fold, *p* = 0.046) and miR-221-5p (5.9-fold, *p* = 0.004) was observed. By day 4 this effect had subsided, with three of the miRNAs—miR-200b-3p, miR-221-5p and miR-205—demonstrating decreased expression in the Ind+S and Ind+A groups when compared to the Ind control. By day 7, the expression of miR-182 had increased again in Ind+A, demonstrating a 2.8-fold increase (Figure 4B).

## 3. Discussion

The molecular mechanisms underlying dental pulp mineralisation and repair are yet to be fully elucidated; however, a deeper understanding would provide exciting opportunities for the development of biologically based, topically applied therapeutic interventions for the stimulation of the natural reparative abilities of the dental pulp. Cellular processes in other parts of the body have already been shown to be regulated, at least in part, by the relationship between miRNAs and other epigenetic modifications [31], and it is likely that DPC mineralisation is no exception.

The effect of SAHA and 5-AZA-CdR on DPC growth rate was determined by Trypan Blue staining and cell counting at a range of time points from days 1 through 14. Notably, the growth curve of DPCs in the NInd group was similar to previously established growth curves of rodent DPCs cultured under similar conditions [5]. There appeared to be minimal change in cell number between the four groups on day 1. Thereafter, culturing the cells in mineralising medium had a positive effect on DPC proliferation, while supplementary addition of both SAHA and 5-AZA-CdR attenuated cell growth. Cells exposed to SAHA displayed a cell growth profile similar to that of those cultured in normal medium, demonstrating an increase in cell number throughout the 14 days, while 5-AZA-CdR appeared to suppress cell proliferation, with no discernible change in the number of cells throughout the duration of the timecourse. Consequently, it can be concluded that, at a concentration of 1μM, both SAHA and 5-AZA-CdR suppress the growth rate of mineralising DPCs, albeit to varying degrees.

Alizarin Red S staining was used to visualise and quantify extracellular calcific nodule formation in cultures. Cultures of the NInd control did not display any red stained deposits indicative of mineralisation, as opposed to the three experimental groups, which all demonstrated strong staining visible from day 11 and increasing in intensity by day 14. At day 14 in particular, SAHA and 5-AZA-CdR both had a positive effect on mineralisation of DPCs compared to mineralising medium alone. This is consistent with previous studies, which demonstrated that short-term treatment with SAHA promoted mineralisation of rodent DPCs [6]. 5-AZA-CdR was also previously shown to accelerate hDPC mineralization [4]. After seven days of culture in mineralising medium and exposure to 5-AZA-CdR, there was an increase in expression of the odontogenic markers DSPP and DMP1. The transcription factors RUNX2, DLX5, and OSX were also upregulated by day 7, as was the formation of mineralised nodules, as detected by Alizarin Red S staining at 14 days [4].

The miRNA expression profile of mineralising DPCs has previously been investigated using microarray analysis to investigate differentially expressed lncRNAs, miRNAs, and mRNAs in human dental pulp stem cells (hDPSCs), which had been induced to differentiate in vitro [32]. The resulting data were used to construct regulatory networks in differentiating DPSCs. Another study used a microarray platform to identify differentially regulated miRNAs in human DPCs, which had been cultured in mineralising medium for 14 days [13]. In that study, it was shown that 22 known mature miRNAs were differentially regulated, of which 10 were downregulated, and 12 were upregulated. In the current study, small RNAseq was used to investigate the miRNA expression profile of rodent DPCs, and no differentially expressed miRNAs were found in mineralising DPCs compared to non-mineralising when using the moderated *t*-test combined with Storey’s bootstrapping method. Subsequently, the less stringent Mann-Whitney U test was applied, which demonstrated 43 miRNAs to be differentially regulated in DPCs cultured in mineralising medium compared to DPCs cultured in normal medium, of which 20 were upregulated and 23 were downregulated. The larger number of differentially expressed miRNAs reported in the present study when using a Mann-Whitney U test compared to the microarray analysis [13] is possibly due to the use of rodent DPCs, rather than human DPCs. Alternatively, it may reflect the use of RNAseq, which can accurately detect a higher percentage of differentially expressed genes compared to microarray analysis, with an increased sensitivity for genes with relatively low expression levels [33,34].

The number of differentially expressed miRNAs (>1.5 fold) in the Ind+S and Ind+A groups compared to the NInd control were similar to the number in the Ind+A group compared to the Ind control, ranging from 24 to 30. The number of differentially expressed miRNAs in Ind+S compared to Ind was notably higher, at 95, which suggests that miRNA expression may be linked to epigenetic modifications, such as histone acetylation [35]. Notably, the majority of differentially regulated miRNAs were upregulated by the epigenetic inhibitors. The magnitude of the fold change was also interesting, with the largest absolute fold change in Ind+S compared to NInd being 8.5, which related to a decrease in the expression of miR-3553. In contrast, the highest fold change in expression in Ind+A compared to NInd was a 603.9-fold increase (miR-881-3p), implying an important role for 5-AZA-CdR in regulating the expression of miR-881-3p, in addition to a number of other miRNAs that were highly differentially expressed.

The results of the RNAseq analysis revealed a number of differentially regulated miRNAs in the experimental groups which have previously been shown to have links with mineralisation and epigenetic processes. miR-205, which was upregulated in Ind+A compared to Ind, has been suggested to inhibit osteogenesis of bone marrow mesenchymal stem cells (BMMSCs) by targeting Runx2, a key transcription factor in osteogenesis [29,30,36]. In addition, miR-205 has been shown to be regulated by DNA methylation in numerous cancers [37,38,39]. Furthermore, many of the miRNAs upregulated in Ind+S compared to the Ind control have been shown to be induced by HDACis, including miR-139-5p [40], miR-375-3p [41], and miR-211-5p [42]. The differential expression of selected miRNAs in response to alterations in histone acetylation and DNA methylation highlights an obvious interaction between miRNAs and other epigenetic modifications. These interactions may be responsible for the range of cellular effects, including the increase in mineralisation, observed in DPCs following exposure to SAHA [6] and 5-AZA-CdR [4].

Potential target genes with a wide variety of functions were identified for the differentially expressed miRNAs in each group. These include Smad4, which acts as a key mediator of the TGF-β/BMP signalling pathway and has been proposed to play a critical role in tooth development [43], and Tnf-α, which encodes a pro-inflammatory cytokine and is upregulated in diseased pulp compared to healthy pulp [44]. In addition, exposure to Tnf-α has been shown to induce mineralisation of DPCs, as evidenced by increased mineralised nodule formation and expression of dentin sialoprotein (DSP), DMP1, and osteocalcin (OC) [45]. Functional analysis of the predicted target genes was carried out by GO enrichment and KEGG pathway analysis. Mineralisation and stem cell-related terms were highly enriched for the target genes, including terms such as regulation of bone mineralisation, Wnt, and MAPK signalling pathways. Notably, similar findings were previously reported when the miRNA expression profile of mineralising DPCs was investigated using microarray profiling [13]. Indeed, the MAPK pathway has been strongly implicated in the regulation of odontoblast activation in response to dentine matrix protein and growth factor stimulants [46].

The expression of miR-346 and miR-881-3p was quantified to validate the RNAseq data. These miRNAs demonstrated the highest fold change in expression in Ind+S and Ind+A compared to Ind, respectively—a finding confirmed by qRT-PCR analysis. The expression of four additional miRNAs—miR-182, miR-200b-3p, miR-221-5p, and miR-205—was also quantified using qRT-PCR. When compared to the NInd group, mineralising medium alone (Ind) did not have a significant effect on the expression of any of the four miRNAs at day 1; however, by day 4, the expression of each miRNA was increased and indeed sustained through day 7 of culture. These data would suggest that inducing DPCs to mineralise does not immediately affect the expression of the selected miRNAs; however, the eventual alteration in expression is ultimately sustained. It is unclear whether this sustained effect is due to continuous replenishment of the mineralising medium, or if inducing DPCs to mineralise results in permanent changes in miRNA expression due to alterations in regulatory mechanisms. In this regard, it would be of interest to quantify the expression of these miRNAs at later time points.

In contrast to the effect of mineralising medium alone, both SAHA and 5-AZA-CdR appeared to have immediate, yet transient effects on miRNA expression when compared to Ind. Both epigenetic inhibitors exerted a strong positive influence on the expression of miR-182 when compared to Ind at day 1; however, by day 4 of culture, this effect had rapidly diminished. By day 7, the expression of miR-182 was increased in Ind+A again; however, this may be due to the reapplication of 5-AZA-CdR to the medium at day 4. Similarly, miR-221-5p was significantly upregulated in Ind+A compared to Ind at day 1, although, by day 4, it demonstrated no significant difference in expression. While the epigenetic inhibitors did not have an effect on miRNA expression in all conditions, when there was an effect, it occurred at day 1, but it had diminished by day 4. This would suggest that, at least for the four miRNAs that were investigated, any effect of epigenetic inhibitors on their expression was immediate, yet brief, with the effects abating following inhibitor withdrawal from the culture. Interestingly, the previous literature has established that the effect of HDACis on miRNA expression is immediate, with exposure to the HDACi LAQ824 resulting in rapid changes in expression of 27 of 67 predetermined miRNAs [47]. In a separate study, the inhibitory effects of the Class II-specific HDACi MC1568 on HDAC activity were demonstrated to diminish following treatment withdrawal, with HDAC activity being restored and histone acetylation levels decreasing accordingly [48]. Similar results have been demonstrated for 5-AZA-CdR, with re-methylation of sites demonstrated to occur following withdrawal of treatment [49].

RNAseq is a powerful and robust tool for the analysis of differential miRNA expression profiles in experimental cultures. In the current study, the use of RNAseq and subsequent bioinformatic analyses, such as GO enrichment and pathway analysis, provided valuable insights into the mechanisms by which miRNAs may regulate DPC mineralisation. Of particular interest was the relationship between miRNAs and other epigenetic machinery in mineralising DPCs, an area of research which has not been previously investigated using high-throughput methods.

While further experimental work is required to validate many of the target genes and pathways identified in the current study, this study has nevertheless confirmed that there is a large-scale alteration in miRNA expression in mineralising DPCs. In addition, it was shown that exposing the cells to pharmacological epigenetic inhibitors results in a much greater change in miRNA expression, suggesting that the increase in mineralisation observed when DPCs are exposed to epigenetic inhibitors may be regulated, at least in part, by the interactions between miRNAs and epigenetic modifiers. Perhaps most importantly, the work carried out in this study enabled the selection of a number of miRNAs for further investigation, which may play a key role in epigenetic regulation of dentine–pulp repair processes. As an overarching aim of regenerative endodontics is to identify biologically-based therapeutic targets for use in treatment of pulpitis, future research should establish an expression profile for the human orthologs of the identified miRNAs in human DPCs in vitro, followed by the use of miRNA mimics and inhibitors to determine the role of these miRNAs in dental pulp reparative processes.

## 4. Materials and Methods

### 4.1. DPC Isolation and Culture

Rodent DPCs were isolated using an enzymatic digestion technique, as previously described [50], from the extracted incisors of freshly sacrificed adult male Wistar Hannover rats, approximately four weeks old and weighing 100–120 g. The rats had been housed in Trinity Biomedical Sciences Institute (TBSI) Animal Facility, Trinity College Dublin under conditions in line with EU guidelines, and had been sacrificed no more than one hour previously by cervical dislocation. Following extraction, the pulp was extirpated from each tooth and transferred to a sterile glass slide, prior to being physically minced into 1 mm^3^ pieces using a sterile scalpel. The pieces were transferred to a 50 mL centrifuge tube (Abdos Labtech, New Delhi, India), containing 4 mL Trypsin-EDTA (Sigma-Aldrich, Arklow, Ireland), prior to incubation at 37 °C, 5% CO_2_ for 40 min (MCO 18-AC, Sanyo, Osaka, Japan), with agitation every 10 min using a pipette to improve dissociation. At the end of the digestion period, the reaction was halted with the addition of normal (non-mineralising) cell culture medium, which consisted of 4 mL α-MEM (Biosera, Labtech International, East Sussex, UK), supplemented with 1% Penicillin/Streptomycin (100 units/mL of penicillin with 100μg/mL streptomycin) (Sigma-Aldrich, Arklow, Ireland) and 10% (*v*/*v*) Foetal Calf Serum (FCS) (Biosera, Labtech International, East Sussex, UK). The cell suspension was passed through a 70 μm cell strainer (Falcon, Corning, Flintshire, UK) into a sterile 50 mL centrifuge tube, which was subsequently centrifuged for 3 min at 250× *g* (Universal 320, Hettich, Tuttlingen, Germany). The pellet was then resuspended in 700 µL normal medium. The cells were seeded in 700 µL in T25 flasks (Sarstedt, Leicester, UK), with each flask representing pulp tissue from four incisor teeth. The cells were expanded in culture until passage 2 for use in subsequent experiments.

### 4.2. Induction of Mineralisation in Experimental Cultures with Addition of Pharmacological Epigenetic Inhibitors

When cultures had reached 80% confluence in six-well plates, as determined by phase contrast microscopy (experimental day 0), wells were assigned to one of the four experimental groups, and the media were changed according to Table 6. For induction of mineralisation, cells were cultured in mineralising medium, which consisted of α-MEM supplemented with 1% Penicillin/Streptomycin and 10% (*v*/*v*) FCS, as described above, with the addition of 50 mg/l ascorbic acid (Sigma-Aldrich, Arklow, Ireland), 10 nM dexamethasone, and 10 mM β-glycerophosphate (Sigma-Aldrich, Arklow, Ireland) [51]. For treatment with a HDACi, 5 mg of SAHA (Sigma-Aldrich, Arklow, Ireland) was dissolved in 3.8 mL dimethyl sulfoxide (DMSO) (Sigma-Aldrich, Arklow, Ireland) to prepare a 5 mM stock solution, prior to supplementation in culture medium at a final concentration of 1 µM [6]. For treatment with a DNMTi, 5 mg of 5-AZA-CdR (Sigma-Aldrich, Arklow, Ireland) was dissolved in 219 μL DMSO to prepare a 100 mM stock solution, prior to supplementation in cell culture medium at a final concentration of 1 µM [4].

The media in all wells were replaced every three days. SAHA was added only on day 0 and was not reapplied throughout the remainder of the experiment, as studies have previously shown that this results in optimal stimulatory effects [6], while 5-AZA-CdR was added on days 0 and 4 [4]. The DNMTi was reapplied due to its instability in aqueous solution and the reversible nature of DNA methylation, with the previous literature indicating that hypo-methylation of DNA reverts following withdrawal of the inhibitor [49].

### 4.3. Cell Growth Curve Analysis

The rate of cell growth over time was investigated by staining the cells with Trypan Blue and counting using a haemocytometer on days 1, 4, 7, 11, and 14. To calculate cell numbers, the medium was carefully aspirated, and the cells were washed briefly with 1 mL Trypsin-EDTA, which had been previously incubated at 37 °C, to remove residual traces of FCS. The initial Trypsin-EDTA wash was aspirated, and an additional 1 mL of Trypsin-EDTA was added. Cultures were then incubated at 37 °C for 5–10 min, or until the cells had detached from the base of the flask, as viewed under phase contrast microscopy, prior to the addition of 1 mL of normal medium to stop the reaction. The cells were then transferred to a single 50 mL centrifuge tube and centrifuged at 100× *g* for 3 min. The supernatant was carefully removed from the cell pellet following centrifugation, and the pellet was resuspended in 1 mL normal medium. A 10 µL volume of this suspension was then added to 10 µL Trypan Blue stain (Sigma-Aldrich, Arklow, Ireland), mixed thoroughly, and incubated for 5 min at room temperature. A sterile glass coverslip was placed on a Bright Line haemocytometer (Hausser Scientific, Horsham, PA, USA), and 10 µL of the Trypan Blue/cell solution were applied with a pipette under the coverslip until it had been completely filled by capillary action. The number of viable cells in each of the four outer squares of the haemocytometer were then counted under phase contrast microscopy. Cells that had excluded the Trypan Blue dye were considered viable. The cell concentration was then calculated as follows:

Cell concentration = average cell count per square × 2 (dilution factor) × 10^4^ (volume of outer square)

Three biological replicates were analysed in duplicate.

### 4.4. Alizarin Red S Staining

To confirm induction of mineralisation, DPCs were stained with 2% (*w*/*v*) Alizarin Red S (Millipore, Cork, Ireland) on days 11 and 14 of culture. Briefly, the medium was aspirated, and the cells were washed with PBS. The cells were then fixed with 1 mL neutral buffered 10% formalin (Sigma-Aldrich, Arklow, Ireland) and washed three times for 5 min each with ultrapure water before being incubated with 1 mL Alizarin Red S solution (2% *w*/*v*) for 20 min. The cells were then washed three times with ultrapure, autoclaved water for 5 min each. Images of the wells were taken and stored as JPEG files, prior to solubilisation of the stain and quantification by spectrophotometric analysis at 405 nm (Tecan Genios Spectrophotometer, Unitech, Dublin, Ireland) [52]. Mineral production per cell was subsequently calculated by dividing the concentration of Alizarin Red S in each group by the corresponding number of cells counted in parallel DPC cultures using a haemocytometer and Trypan Blue staining, as described above. Three biological replicates were analysed in triplicate.

### 4.5. miRNA Extraction

miRNA was isolated and extracted as a component of the total RNA sample using the miRNeasy Mini Kit (Qiagen, Redwood City, CA, USA), according to the manufacturer’s instructions. Extracted RNA was analysed using a Nanodrop 2000 c spectrophotometer (Thermo-Fisher, Dublin, Ireland) to determine the yield and quality. RNA samples were considered to be of acceptable quality for RNAseq and qRT-PCR if the 260/280 and 260/230 ratios were greater than 1.8 and 2.0, respectively.

### 4.6. RNA Sequencing

RNA, including miRNA, which had been isolated from each of the four experimental groups on day 4 of culture, was analysed by GeneWiz (GeneWiz, South Plainfield, NJ, USA). Three technical replicates were analysed for each group. Library preparation was carried out using the TruSeq Small RNA Library Prep Kit (Illumina, San Diego, CA, USA), prior to RNA sequencing using the Illumina HiSeq 2500 system (Illumina, San Diego, CA, USA). Data were supplied as 12 fastq files, which were analysed using Strand NGS ver3.4 (Strand Life Sciences, Bengaluru, India). The reads were initially trimmed to remove adapter sequences (CutAdapt 1.14, Python 2.7.10), following which sequence reads 15 to 35 nucleotides in length were retained for subsequent analysis. The reads were then aligned against the rat reference genome from assembly Rnor_6.0 (RGSC, 2014) [53]. To identify the various sRNA classes, reads were searched against tRNAscan-SE [18] to annotate tRNAs, miRBase-21 [20] to annotate miRNAs, and Ensembl (release 98, [19]) to annotate the remaining small RNAs, such as ribosomal RNA and small nucleolar RNA (snoRNA). Prior to analysis, miRNA counts were normalised to reads per kilobase of transcript, as well as per million mapped reads (RPKM).

### 4.7. Identification of Differentially Expressed miRNAs

For characterisation of miRNA expression profiles under the various experimental conditions, the data were organised such that NInd was used as a control against which Ind, Ind+S, and Ind+A could be compared, and Ind was used as a control against which Ind+S and Ind+A could be compared, resulting in a total of five experimental vs. control pairings. Differential expression was performed on each pairing on Strand NGS, using both a Mann-Whitney U test (fold change cut-off >1.5, *p* ≤ 0.05) and a moderated *t*-test (fold change cut-off >1.5, *q* ≤ 0.05), independently of each other.

### 4.8. Functional Annotation

A list of predicted target genes for each list of differentially regulated miRNAs was generated using Strand NGS, which utilises the miRDB database for target gene prediction. A *p*-value cut-off of ≤0.05 was applied. GO enrichment, using DAVID v6.8 (https://david.ncifcrf.gov/, accessed on 16 October 2020) [22,23] and KEGG pathway analysis, using DIANA-miRPath v.3 (http://www.microrna.gr/miRPathv3/, accessed on 22 October 2020) [54], were carried out on the resulting lists of target genes.

### 4.9. Selection of miRNAs for Validation Using qRT-PCR

The mature miRNA sequences which were differentially regulated in Ind+S and Ind+A compared to NInd, as determined by a moderated *t*-test, were compiled into one datasheet consisting of 90 mature miRNA sequences. A Venn diagram was constructed, which displayed the number of shared differentially expressed miRNAs between groups. The same process was applied to mature miRNAs which were differentially regulated in Ind+S vs. Ind and Ind+A vs. Ind, yielding a total of 118 mature miRNA sequences. A selection of miRNAs were identified from the datasets through a search of the literature, using PubMed (https://pubmed.ncbi.nlm.nih.gov/, accessed on 12 April 2021) to identify previous associations with mineralisation, differentiation, or epigenetic processes. The list of miRNAs was further refined based on the results of in-depth functional analysis, as described below.

#### 4.9.1. Target Gene Prediction

For each of the mature miRNA sequences, three different software tools were used to generate a list of predicted target genes—TargetScan (http://www.targetscan.org/vert_80/, accessed on 8 October 2020) [55], miRDB (http://mirdb.org/, accessed on 8 October 2020) [56,57], and DIANA-microT-CDS (http://www.microrna.gr/webServer/, accessed on 8 October 2020) [58]. To increase specificity, only target genes which were predicted by at least two of the target prediction tools were considered for further analysis [59,60,61].

#### 4.9.2. Functional Annotation of Individual miRNAs

GO enrichment and KEGG pathway analysis was carried out for each list of target genes using DAVID bioinformatics resources. GO terms and KEGG pathways that were related to stem cell or mineralisation processes were noted for further investigation.

#### 4.9.3. Orthology Analysis

The existence of a human equivalent for each miRNA was determined by searching miRBase for the miRNA, replacing the prefix ‘rno’ with ‘hsa’. If present, the sequence of the mature human miRNA was manually aligned and compared to that of the equivalent rodent miRNA using the Basic Local Alignment Search Tool (BLAST) software (https://blast.ncbi.nlm.nih.gov/Blast.cgi, accessed on 12 April 2021) [62]. The target genes of the human miRNA were then predicted using TargetScan, miRDB and DIANA-microT-CDS, as described above. The list of target genes was compared to that of the rodent miRNA, and the shared target genes were used to generate a functional analysis profile through GO enrichment and KEGG pathway analysis, so as to determine shared functions of the human and rodent miRNA. Using the predicted target genes, functional annotation, and conservation in humans as references, a selection of miRNAs—miR-182, miR-200b-3p, miR-205, miR-221-5p, miR-346, and miR-881-3p—were chosen for quantification using qRT-PCR.

### 4.10. Validation of miRNA Expression Using qRT-PCR

The expression of the selected miRNAs was further investigated by quantification using qRT-PCR. Total RNA, containing miRNA, which had been isolated on days 1, 4, and 7 of culture, was converted into cDNA using the miRCURY LNA RT Kit (Qiagen, Redwood City, CA, USA), following the manufacturer’s instructions. Subsequently, qRT-PCR for quantification of the selected miRNAs was carried out using the miRCURY LNA SYBR Green PCR Kit (Qiagen, Redwood City, CA, USA) and associated primers (miRCURY LNA miRNA PCR Assay, Qiagen, Redwood City, CA, USA). Primer sequences are listed in Table 7. *U6* was used as an internal reference gene. Reactions were performed using the Applied Biosystems 7500 Fast Real-Time PCR System (Thermo-Fisher, Dublin, Ireland) and consisted of an initial incubation at 95 °C for 2 min to activate the PCR reaction, followed by 40 repeated amplification cycles, with a typical cycle involving denaturation at 95 °C for 10 s followed by combined annealing and extension at 56 °C for 60 s. Three independent biological replicates were analysed in duplicate. Fold change in expression was calculated using the formula 2^−(ΔCt(Exp)−ΔCt(Ctrl))^, where Ct is the threshold cycle, ΔCt is Ct(target gene) − Ct(reference gene), Exp is the experimental and Ctrl is the control samples. Three independent biological replicates were analysed in duplicate.

### 4.11. Statistical Analysis

One-way analysis of variance (ANOVA) combined with post-hoc Tukey’s test was used to analyse the effect of four different culture conditions on cell proliferation, metabolic activity, and mineralisation. Strand NGS software was used to carry out statistical analysis on the RNAseq data, including differential expression analysis, for which the moderated *t*-test combined with Storey’s bootstrapping to account for multiple testing was used. Mean and standard deviation were used for statistical analysis. Differences were considered statistically significant at *p* ≤ 0.05. For analysis of qRT-PCR data, an independent *t*-test was applied to the fold change values.

## Figures and Tables

**Figure 1 ijms-24-08631-f001:**
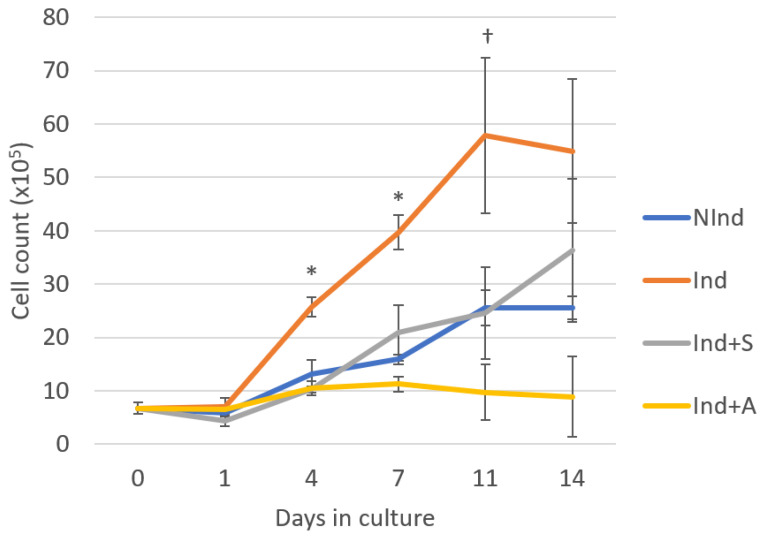
Timecourse study of the growth rate of rodent dental pulp cells (DPCs), as determined by Trypan Blue staining and cell counting. DPCs cultured in mineralising medium (Ind) demonstrated greater cell growth from days 4 through 14 compared to the other three groups, including the non-mineralising control (NInd), while SAHA (Ind+S) and 5-AZA-CdR (Ind+A) appeared to suppress this effect. One-way analysis of variance (ANOVA), followed by Tukey’s post hoc test, were used to analyse statistical significance. All charted data are represented as means ± standard errors of the mean (SEMs). All time points and groups based on three biological replicates carried out in duplicate. * Statistically significant difference compared to NInd, Ind+S and Ind+A, *p* ≤ 0.05. † Statistically significant difference compared to Ind+A, *p* ≤ 0.05.

**Figure 2 ijms-24-08631-f002:**
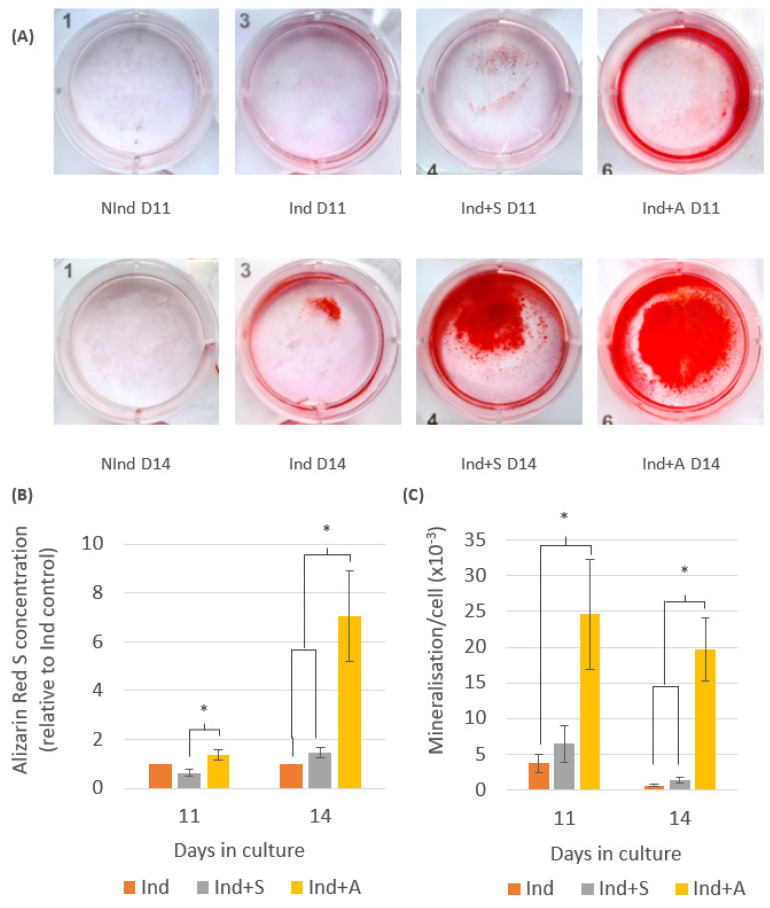
Analysis of mineralisation in rodent DPCs in the four experimental groups: NInd, Ind, Ind+S, and Ind+A. (**A**) Representative staining of calcified deposits in the control (NInd) and experimental (Ind, Ind+S and Ind+A) groups with Alizarin Red S after 11 and 14 days in culture. (**B**) Quantification of Alizarin Red S stain in rodent DPCs. By day 11, 5-aza-2′-deoxycytidine (5-AZA-CdR) had a positive effect on mineralisation, with this effect being significantly different from Ind+S at day 11 and 14, as well as from Ind at day 14. (**C**) Mineralisation per cell analysis in rodent DPCs. By day 11, both 5-AZA-CdR and suberoylanilide hydroxamic acid (SAHA) had a positive effect on mineralisation compared to mineralising medium alone (Ind), with this effect being significant in Ind+A on day 11 and 14. On day 14, mineralisation per cell was also significantly higher in Ind+A compared to Ind+S. One-way ANOVA, followed by Tukey’s post hoc test, were used to analyse statistical significance between groups. All charted data are represented as means ± SEMs. Data are representative of three biological replicates carried out in triplicate. * Statistically significant difference between indicated groups, *p* ≤ 0.05.

**Figure 3 ijms-24-08631-f003:**
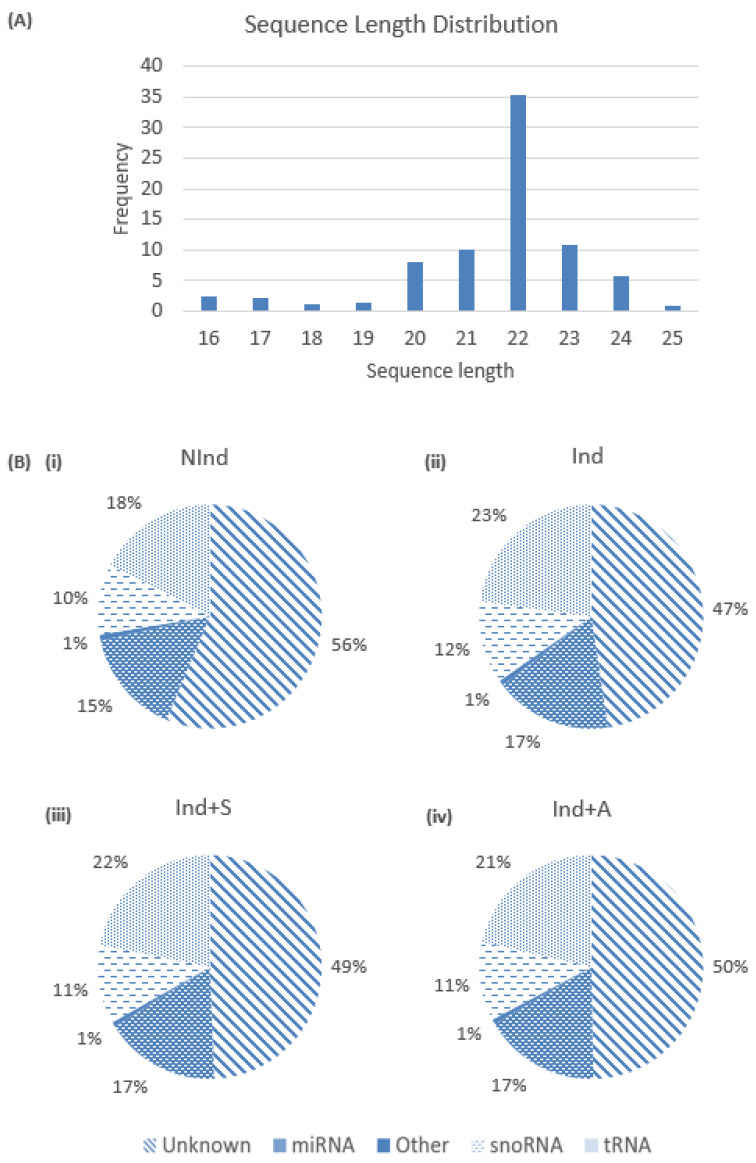
Distribution of small RNA reads and classes in DPC cultures. (**A**) Read length distribution in DPCs across the four experimental groups. (**B**) Pie charts showing the various small RNA classes in (**i**) NInd, (**ii**) Ind, (**iii**) Ind+S, and (**iv**) Ind+A. ‘Other’ is comprised of small cajal body-specific RNA (scaRNA) and small nuclear RNA (snRNA). tRNA = transfer RNA, snoRNA = small nucleolar RNA.

**Figure 4 ijms-24-08631-f004:**
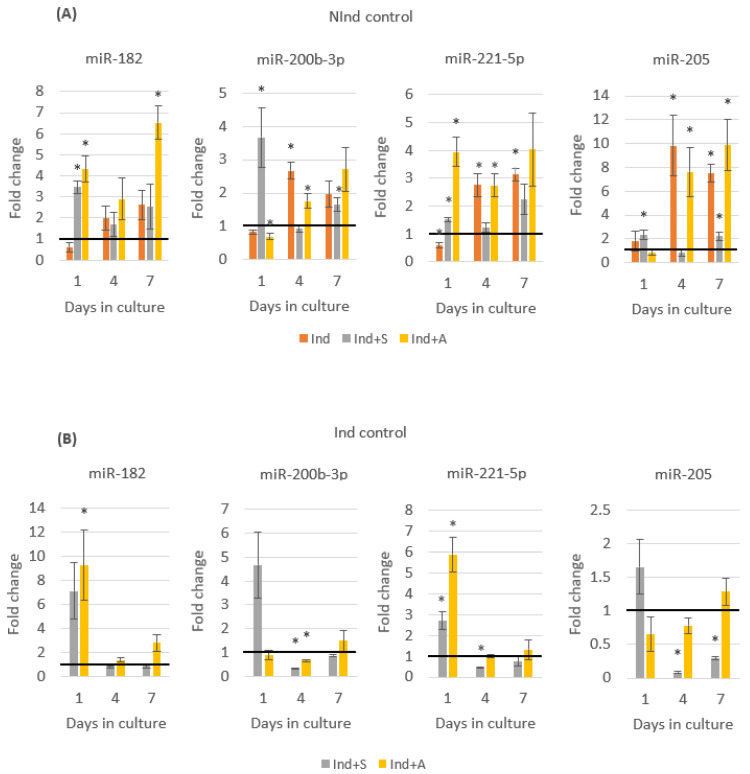
Timecourse analysis, demonstrating a fold change in expression of miR-182, miR-200b-3p, miR-221-5p, and miR-205 relative to (**A**) NInd and (**B**) Ind control. * Statistically significantly different compared to control, *p* ≤ 0.05. An independent *t*-test was used to analyse statistical significance between groups at each time point. All charted data are represented as mean ± SEM. Data are representative of three biological replicates carried out in duplicate. Black horizontal line indicates no change in expression.

**Table 1 ijms-24-08631-t001:** The number of differentially expressed miRNAs in each experimental group compared to NInd and Ind controls, as determined by moderated *t*-test and Storey’s bootstrapping method.

Experimental Group	Upregulated	Downregulated	Total
Compared to NInd			
Ind	0	0	0
Ind+S	15	11	26
Ind+A	29	1	30
Compared to Ind			
Ind+S	62	33	95
Ind+A	23	1	24

**Table 2 ijms-24-08631-t002:** The number of differentially expressed miRNAs and their target genes for each experimental pairing. For each pairing, a selection of target genes related to stem cell differentiation, mineralisation or epigenetic processes is included. Alpl = alkaline phosphatase, biomineralization associated, Calb1 = calbindin 1, Dkk1 = dickkopf WNT signaling pathway inhibitor 1, DEmiRNAs = differentially expressed miRNAs, Map2k1 = mitogen-activated protein kinase kinase 1, Map3k8 = mitogen-activated protein kinase kinase kinase 8, Mapkapk2 = MAPK activated protein kinase 2, Mmp13 = matrix metallopeptidase 13, Sfrp4 = secreted frizzled related protein 4, Smad1 = SMAD family member 1, Sox4 = SRY-box transcription factor 4, Tgif2 = TGFB induced factor homeobox 2, Wisp3 = WNT1 inducible signaling pathway protein 3, Wnt2b = Wnt family member 2b.

Experimental Group	No. of DEmiRNAs	No. of Target Genes	Selected Target Genes of Interest
Compared to NInd			
Ind+S	26	150	*Wnt2b*, *Map2k1*
Ind+A	30	468	*Alpl*, *Tgif2*, *Dkk1*, *Mmp13*, *Smad1*, *Wnt10b*
Compared to Ind			
Ind+S	95	425	*Sfrp4*, *Wnt2b*, *Map3k8*, *Smad7*, *Sox4*, *Tgif1*
Ind+A	24	345	*Dkk1*, *Mapkapk2*, *Smad1*, *Tgif2*, *Wisp3*, *Calb1*

**Table 3 ijms-24-08631-t003:** Gene ontology (GO) terms relevant to mineralisation and stem cell differentiation processes associated with the target genes in each experimental pairing, as determined by the Database for Annotation, Visualisation and Integrated Discovery (DAVID).

Experimental Group	GO Term	GO ID	No. of Genes
Compared to NInd			
Ind+S	Regulation of stress-activated MAPK cascade	GO:0032872	5
Ind+A	Ossification	GO:0001503	26
	Osteoblast differentiation	GO:0001649	16
	Biomineral tissue development	GO:0031214	12
	Tooth mineralisation	GO:0034505	5
Compared to Ind			
Ind+S	Regulation of MAPK cascade	GO:0043408	22
	Stem cell proliferation	GO:0072089	6
Ind+A	Osteoblast differentiation	GO:0001649	8
	Bone mineralisation	GO:0030282	5
	MAPK cascade	GO:0000165	18

**Table 4 ijms-24-08631-t004:** Kyoto Encyclopedia of Genes and Genomes (KEGG) pathways relevant to mineralisation and stem cell differentiation processes associated with the target genes in each experimental pairing, as determined by DIANA-miRPath v.3.

Experimental Group	KEGG Pathway	KEGG ID	No. of Genes
Compared to NInd			
Ind+S	TGF-β signalling pathway	rno04350	21
	Wnt signalling pathway	rno04310	32
Ind+A	TGF-β signalling pathway	rno04350	17
	MAPK signalling pathway	rno04010	52
Compared to Ind			
Ind+S	MAPK signalling pathway	rno04010	95
	Wnt signalling pathway	rno04310	57
	TGF-β signalling pathway	rno04350	36
Ind+A	TGF-β signalling pathway	rno04350	17
	MAPK signalling pathway	rno04010	40

**Table 5 ijms-24-08631-t005:** Fold change in expression of miR-346 in Ind+S compared to Ind and miR-881-3p in Ind+A compared to Ind, as determined by RNAseq and qRT-PCR.

miRNA	Experimental Group (Compared to Ind)	Fold Change (RNAseq)	Fold Change (qRT-PCR)
miR-346	Ind+S	9.03	3.65
miR-881-3p	Ind+A	273.97	125.31

**Table 6 ijms-24-08631-t006:** The culture conditions of each experimental group.

Group	Medium
Non-Induced (NInd)	Normal medium
Induced (Ind)	Mineralising medium
Induced + SAHA (Ind+S)	Mineralising medium + 1 µM SAHA
Induced + 5-AZA-CdR (Ind+A)	Mineralising medium + 1 µM 5-AZA-CdR

**Table 7 ijms-24-08631-t007:** Primers used in qRT-PCR analysis. For each primer, the product name, GeneGlobe ID, and primer sequence are provided.

miRNA	Product Name	GeneGlobe ID	Primer Sequence (5′–3′)
miR-182	mmu-miR-182-5p miRCURY LNA miRNA PCR Assay	(YP0020508)	UUUGGCAAUGGUAGAACUCACACCG
miR-200b-3p	rno-miR-200b-3p miRCURY LNA miRNA PCR Assay	(YP00205111)	UAAUACUGCCUGGUAAUGAUGAC
miR-205	rno-miR-205-5p miRCURY LNA miRNA PCR Assay	(YP00205958)	UCCUUCAUUCCACCGGAGUCUGU
miR-221-5p	rno-miR-221-5p miRCURY LNA miRNA PCR Assay	(YP02116701)	ACCUGGCAUACAAUGUAGAUUUC
miR-346	rno-miR-346 miRCURY LNA miRNA PCR Assay	(YP00205130)	UGUCUGCCUGAGUGCCUGCCUCU
miR-881-3p	rno-miR-881-3p miRCURY LNA miRNA PCR Assay	(YP00205578)	UAACUGUGGCAUUUCUGAAUAG
*U6*	U6 snRNA miRCURY LNA miRNA PCR Assay	(YP00203907)	GUGCUCGCUUCGGCAGCACAUAUACUAAAAUUGGAACGAUACAGAGAAGAUUAGCAUGGCCCCUGCGCAAGGAUGACACGCAAAUUCGUGAAGCGUUCCAUAUUUUU

## Data Availability

The data discussed in this publication have been deposited in NCBI’s Gene Expression Omnibus [63] and are accessible through GEO Series accession number GSE229197 (https://www.ncbi.nlm.nih.gov/geo/query/acc.cgi?acc=GSE22919.7, accessed on 12 April 2021).

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
