# Peer review of "Characterisation of miRNA Expression in Dental Pulp Cells during Epigenetically-Driven Reparative Processes"

_ijms, 2023, doi:10.3390/ijms24108631_

Round 1

Reviewer 1 Report

In the Manuscript authors investigated effects of epigenetic factors on odontogenic differentiation of dental pulp cells (DPCs), and consequently miRNA expression. Manuscript is adequately conceptualized and well written.

Since experiments were conducted on rodent DPCs, Ethical approval is necessary. Also, handling of the animals, their welfare and sacrifice method should be described. Number of extracted teeth, and consequently number of pulp tissue used for DPCs isolation must be specified.

In Alizarin red staining protocol, it is not clear how the ‘’mineral production per cell was subsequently calculated based on parallel cell counting assays using Trypan Blue staining’’. How did Trypan Blue staining affect the calculation of mineral production?

Author Response

Thank you for your constructive comments. Changes have been made to the manuscript in response to the suggestions as outlined below:

Point 1: Ethical approval is necessary

Response: An ethics statement has been added to the back matter of the manuscript.

Point 2: Also, handling of the animals, their welfare and sacrifice method should be described.

Response: The housing conditions and method of sacrifice of the animals has been added to the methods section.

Point 3: Number of extracted teeth, and consequently number of pulp tissue used for DPCs isolation must be specified.

Response: Number of extracted teeth per flask has been included.

Point 4: In Alizarin red staining protocol, it is not clear how the ‘’mineral production per cell was subsequently calculated based on parallel cell counting assays using Trypan Blue staining’’. How did Trypan Blue staining affect the calculation of mineral production? 

Response: The method of calculating mineralisation per cell by using Trypan Blue staining has been clarified.

Reviewer 2 Report

The manuscript is well-written and tackles a very important topic. The results presented will help creation of new knowledge in the field. My concern is related to excessive use of auto-citations throughout the paper, overvaluing the work of the group itself and ignoring relevant work by others in this field.

Author Response

Thank you for the feedback on this manuscript. Changes have been made as described below:

Point: My concern is related to excessive use of auto-citations throughout the paper, overvaluing the work of the group itself and ignoring relevant work by others in this field.

Response: Changes have been made to the reference list to include other works in the field.

Reviewer 3 Report

I read with much interest the manuscript  entitled "Characterisation of miRNA expression in dental pulp cells during epigenetically-driven reparative processes".

References are correctly included. Materials methods and results are well described. I does not have any observation.

Author Response

Thank you for taking the time to read and review this manuscript.